# Toward Robust Live Streaming over LEO Satellite Constellations: Measurement, Analysis, and Handover-Aware Adaptation

## ABSTRACT

Live streaming has experienced significant growth recently. Yet this rise in popularity contrasts with the reality that a substantial segment of the global population still lacks Internet access. The emergence of Low Earth orbit Satellite Networks (LSNs), such as SpaceX's Starlink and Amazon's Project Kuiper, presents a promising solution to this issue. Nevertheless, our measurement study reveals that existing live streaming platforms may not be able to deliver a smooth viewing experience on LSNs due to frequent satellite handovers, leading to frequent rebuffering events. Current state-of-the-art learning-based Adaptive Bitrate (ABR) algorithms, even when trained on satellite network traces, fail to manage the abrupt network variations associated with these handovers effectively. To address these challenges, for the first time, we introduce Satellite-Aware Rate Adaptation (SARA), a versatile and lightweight middleware that can be seamlessly integrated with various ABR algorithms to enhance the performance of live streaming over LSNs. SARA intelligently modulates video playback speed and furnishes ABR algorithms with key insights derived from the distinctive network characteristics of LSNs, thereby aiding ABR algorithms in making informed bitrate selections and effectively minimizing rebuffering events that occur during satellite handovers. Our extensive evaluation shows that SARA can effectively reduce the rebuffering time by an average of 39.41% and slightly improve latency by 0.65% while only introducing an overall loss in bitrate by 0.13%.

## CCS CONCEPTS

• **Information systems** → **Multimedia streaming**; • **Networks** → **Wireless access networks**.

## KEYWORDS

Multimedia Services, Low Earth Orbit Satellite Network, Network Performance Measurement & Optimization

## 1 INTRODUCTION

Live streaming witnessed a significant 99% growth over the past three years,[1] currently engaging almost 30% of Internet users on

---

[1]https://www.forbes.com/sites/paultassi/2020/05/16/report-livestream-viewership-grew-99-in-lockdown-microsofts-mixer-grew-02/?sh=351f5a8e76cb

*MM'24, Oct 28-Nov 1, Melbourne, Australia*
© 2024 Copyright held by the owner/author(s). Publication rights licensed to ACM.
ACM ISBN 978-x-xxxx-xxxx-x/YY/MM. . . $15.00
https://doi.org/10.1145/nnnnnnn.nnnnnnn

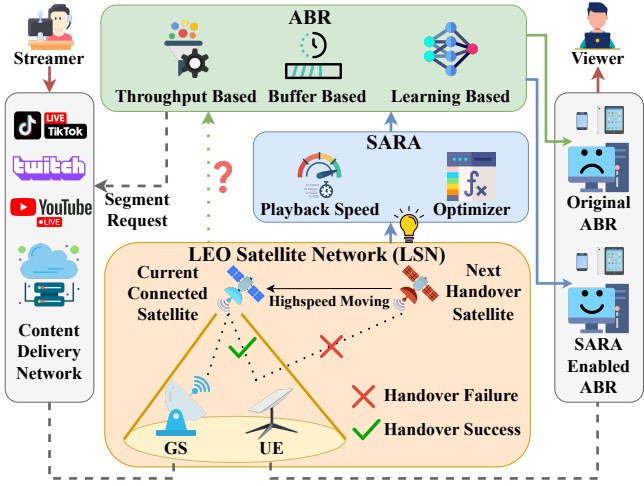

**Figure 1: Overview of live streaming services over LSNs where SARA can be easily integrated with various ABR algorithms to enhance viewers' live streaming experience.**

a weekly basis.[2] While it has become integral to urban life, approximately 34% of the global population remains without Internet access.[3] Fortunately, the commercial success of Low Earth orbit Satellite Networks (LSNs) operators, notably SpaceX's Starlink and Amazon's Project Kuiper [28], presents a promising solution to bridge this gap.

Low Earth Orbit (LEO) satellites, orbiting significantly closer to the Earth than their Geosynchronous Orbit counterparts, offer reduced communication delays but with a much smaller coverage area.[4] Additionally, due to their faster orbital periods of 128 minutes or less, they only have a visible duration of 2-10 minutes at a fixed location on the Earth [2, 24]. Consequently, a constellation of LEO satellites, forming an LSN, is necessary to achieve comprehensive global coverage and provide seamless, high-quality Internet services [22, 40]. With a dense satellite constellation, LSNs can offer significant advantages, especially in areas where establishing terrestrial infrastructure is either challenging or cost-prohibitive. By resorting to LSNs, live streaming can fully unleash its potential in currently underserved areas, reaching a broader audience and offering more resilient services.

Yet, user equipment (UE) connected to LSNs, even stationary ones, inevitably experience frequent satellite handovers due to satellite motion. Recent measurement studies have shown that Starlink updates the UE-satellite link at a granularity of 15 seconds [3, 35]. Such frequent handovers can lead to network outages [17, 27, 41], adversely affecting live streaming services which rely on continuous connectivity. Live streaming services predominantly use HTTP adaptive streaming and HTTP Live Streaming protocol to deliver

---

[2]https://www.statista.com/statistics/1351162/live-streaming-global-reach/
[3]https://www.statista.com/statistics/1229532/
[4]https://earthobservatory.nasa.gov/features/OrbitsCatalog

video content. At the heart of this technology are Adaptive Bitrate (ABR) algorithms. By selecting from various pre-encoded video quality levels, ABR algorithms can combat variations in underlying network conditions to ensure viewers receive high-quality and low latency video streams.

Nevertheless, these ABR algorithms are optimized for terrestrial networks and lack awareness of the sudden network disruptions caused by frequent satellite handovers. According to our measurements, *Twitch's* ABR algorithm struggles with the variable network conditions of LSNs, leading to video rebuffering events at the frequency of several minutes during Internet peak hours. Even using state-of-the-art learning based ABR algorithms trained directly with satellite network data, the streaming quality still struggles as these algorithms do not distinguish LSNs from such wireless terrestrial networks as WiFi or cellular networks, where the handovers are mostly attributable to user movements, leading to sporadic and infrequent occurrences and a more gradual degradation in network conditions. In contrast, LEO satellites usually move at the speed of 28,080 kilometres per hour[5], which is two magnitudes higher than the typical moving speed of pedestrian or vehicular users in wireless terrestrial networks. Making those handovers in LSNs usually occur at a far more frequent pace, and the consequent shifts in network conditions are not only much more abrupt but also typically accompanied with disruptions. Such fast-paced handovers in LSNs, coupled with the abruptness of network condition variations and disruptions, severely challenge the adaptability of current ABR algorithms, outpacing their capacity to adjust and resulting in sub-optimal performance and diminished streaming experiences in the LSN context.

We strive to tackle this challenge and, for the first time, propose a versatile and lightweight middleware solution named Satellite-Aware Rate Adaptation (SARA), which can seamlessly work with various existing ABR algorithms, furnishing them with insightful information interpreted from specific network characteristics unique to LSNs, enhancing their adaptability, and enabling them to more effectively navigate and optimize streaming performance within this brand new type of network environment. The design of SARA is motivated by our preliminary measurements of the Starlink network, which focus specifically on its performance in the context of live streaming services and identifying key characteristics that can significantly impact users' viewing experience. As illustrated in Figure 1, SARA is explicitly crafted to enhance the performance of a variety of different types of ABR algorithms such as Robust Model Predictive Control (RobustMPC), Buffer Based Approach (BBA), and Pensieve [12, 25, 38] for live streaming services over LSNs where SARA can intelligently control the playback speed by both accelerating and decelerating the video and assist the ABR algorithm in selecting the suitable bitrate to avoid rebuffering events during satellite handovers. Our extensive evaluation shows that SARA can effectively reduce rebuffering time by an average of 39.41% while simultaneously maintaining both high bitrate and low latency for live streaming viewers.

Our primary contributions are as follows:

- We conduct a preliminary measurement study focusing on LSNs, using Starlink as a case study. The results, particularly

on *Twitch*, reveal that existing ABR algorithms struggle to provide smooth viewing experiences under LSN conditions, often resulting in frequent video pauses ranging from 5.93 seconds to 23.57 seconds.
- By further in-depth analysis of Starlink's outage, we discover that the occurrence of outages surges by 150% during Internet peak hours compared to non-peak hours.
- Motivated by our measurement study, we propose for the first time a versatile middleware named SARA to dynamically adjust video playback speed and utilize throughput and buffer scalars to guide existing ABR algorithms with bitrate selection and combat network outages under LSNs.
- Our extensive evaluation shows that SARA can significantly reduce average rebuffering time by 39.41% at a negligible cost of an average loss in bitrate of 0.65%.

## 2 MEASUREMENTS AND ANALYSIS

We first conduct a measurement on Starlink to help us understand the network characteristics and challenges associated with LSNs in the context of live streaming services. In this paper, we focus on the client side and evaluate the downlink performance of the Starlink network. Our Starlink UE is positioned at a vantage point to ensure an unobstructed view of the sky with an obstruction ratio of 0.734% as reported by the Starlink portal[6]. We collect our measurement data on clear days to prevent any performance degradation due to extreme weather conditions [24]. The data collection commenced in late March 2023 and spanned approximately four months. During this period, we measured basic network data, including network delay and throughput. Additionally, we collected Starlink outage history using the Starlink mobile app[7], which provided valuable insights into the frequency and duration of network disruptions. Beyond these general network measurements, we also evaluate the performance of *Twtich*, *TikTok LIVE* and *YouTube Live* in order to understand the performance of ABR algorithms under LSNs in the context of live streaming.

## 2.1 Live Streaming Challenges in the LSNs

In this paper, we specifically evaluate the performance of *Twitch*, as our preliminary measurements indicate that all three platforms exhibit comparable performance, and *Twitch* is the predominant live streaming platform in North America[8]. To evaluate *Twitch* ABR's performance, we choose a channel that streams content at a typical resolution of 1920 × 1080, with a default frame rate of 35 FPS and bitrates around 6 Mbps. During our measurement, we observed a stark contrast in the number of rebuffering events and duration between LSN and terrestrial network. In terms of duration, rebuffering lasts an average of 5.93 seconds with a peak of 23.57 seconds for LSN, while for terrestrial networks, the maximum rebuffering time is only 4.64 seconds with an average of 2.97 seconds. In terms of frequency, we observed an average of 1.04 rebuffering events per hour with LSN, with instances of up to 8 rebuffering events occurring within one hour. In comparison, terrestrial networks exhibit

---

[5]https://www.esa.int/ESA_Multimedia/Images/2020/03/Low_Earth_orbit

[6]http://dishy.starlink.com/

[7]https://play.google.com/store/apps/details?id=com.starlink.mobile

[8]https://www.statista.com/statistics/1409393/top-live-streaming-platforms-hours-watched/

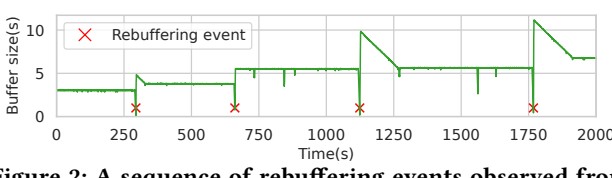

Figure 2: A sequence of rebuffering events observed from *Twitch* on the Starlink network.

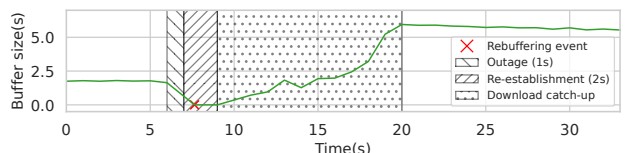

Figure 3: A typical rebuffering event, illustrating that short duration outages can still cause video rebuffering due to connection re-establishment in the application layer.

an average of 0.27 rebuffering events per hour, with no sequential rebuffering events observed within one hour.

Figure 2 provides a 30-minute snapshot depicting buffer size fluctuations on *Twitch*, which highlights the inadequacy of the existing ABR algorithms in LSNs. Notably, the average rebuffering duration, at 5.93 seconds, consistently exceeds the current Latency to Broadcaster (LtB), which is 3 seconds in the figure. LtB represents the time taken for content to travel from the streamer's side to the viewer's side and also serves as the maximum limit for the amount of video content that can be preloaded on the viewer's side. As a result, the buffer is consistently emptied after each network outage. Even when the ABR algorithm always selects the lowest bitrate to minimize rebuffering, completely avoiding such events remains difficult. We also noticed that the outage reported by the Starlink mobile app could lead to an extended rebuffering duration at the application level. Figure 3 zooms in one particular rebuffering event. As shown, there is a 1-second outage followed by a 2-seconds buffer re-establishment, which ultimately leads to a rebuffering event lasting for 1.4 seconds. Therefore, even if the buffer size marginally exceeds the outage duration at the onset of the outage, users may still encounter rebuffering events.

## 2.2 In-Depth Analysis of Network Outage

Numerous studies have confirmed that network outages in LSNs, such as Starlink, are primarily due to satellite handovers [17, 24, 29, 41]. It also has been demonstrated through both end-to-end measurements and signal analysis that Starlink schedules UE-satellite link reallocation every 15 seconds, specifically occurring at the 12th, 27th, 42nd, and 57th seconds of each minute [3, 35]. However, the exact satellite handover impact is still unknown as the handover might succeed or fail, which depends on complex factors such as the satellite load and the number of candidate satellites. To the best of our knowledge, the precise handover impact is still under exploration, and no unified pattern has been identified. Building on this, we delve deeper into the distribution of outage events and utilize statistical models to formalize the frequency and duration of Starlink network outages.

Over three months, we recorded a total of 3, 755 outage events. We consider each outage event as two separate statistical models: one for the occurrence of outages and another for their duration.

Table 1: The top 3 distributions with the lowest SSE

| Distribution | SSE | Mean | Standard Deviation |
|---|---|---|---|
| Normal Inverse Gaussian | 2.84 | 0.78 | 0.13 |
| Weibull | 3.45 | 0.80 | 0.27 |
| Inverted Gamma | 4.28 | 0.23 | 1.95 |

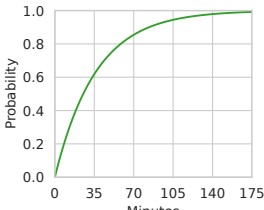 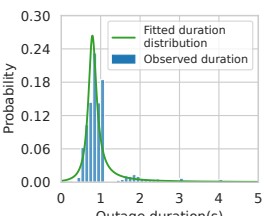

Figure 4: Probability of the first outage over time.

Figure 5: Outage duration distribution.

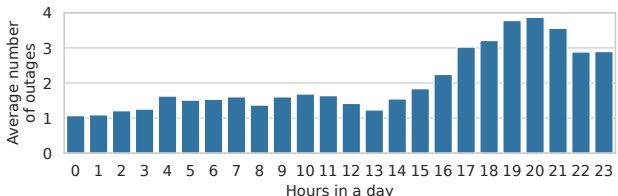

Figure 6: Average number of outages within hours of a day.

We analyze the outage occurrence using a binomial distribution. The Cumulative Density Function (CDF) is illustrated in Figure 4. Our analysis indicates that there is an approximately 80% chance of experiencing an outage within a 60-minute interval. For outage duration distribution, we assess various distributions commonly used for modelling event duration and compare their fitting accuracy based on the sum squared error (SSE), as presented in Table 1. The Normal Inverse Gaussian (NIG) distribution shows the lowest error, with an SSE of 2.84 and a standard deviation of 0.13. Figure 5 illustrates the probability distribution function of different outage durations, comparing the actual outage data (represented by bars) with the NIG distribution model (depicted as a line). The heavy-tail property of the NIG distribution also corresponds well with our observations: while 87.33% outage duration is less than 2 seconds, they could extend beyond 5 seconds, up to a maximum of 31 seconds, with a probability of 2.73%. While the majority of durations are less than 2 seconds, given the connection re-establishment delays discussed in Section 2.1, these short-duration outages can still trigger rebuffering events in live streaming services.

Additionally, we notice a clear correlation between the occurrence and duration of outages and specific times of day. As illustrated in Figure 6, the frequency of outages experiences a dramatic increase from 15:00 to 18:00, peaking at 20:00, which coincides with the Internet rush hour[9]. The duration of these outages varies significantly throughout the day, with the standard deviation reaching its highest at 8.03 around 00:00, then collapsing to 0.77 around 01:00. Given that each Starlink satellite has a finite number of antennas and shares communication channels among four users [29], the likelihood of handover failure escalates due to the increased frequency of handover operations required per satellite during these

---

[9]https://en.wikipedia.org/wiki/Internet_rush_hour

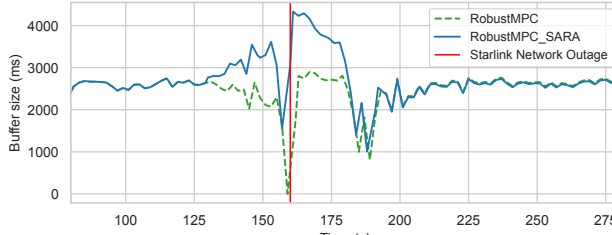

Figure 7: Overview of SARA.

peak times. Intriguingly, our findings show that the evening Internet rush hour is more pronounced compared to the afternoon which suggests that the majority of Starlink users are more likely to engage in entertainment activities, such as live streaming.

## 3 SYSTEM MECHANISM

### 3.1 SARA Overview

From our measurement and analysis, we develop SARA, a versatile middleware that can be easily integrated with a variety of existing ABR algorithms, whether they are buffer based, throughput based, or learning based. SARA's principal aim is to optimize user QoE by effectively minimizing rebuffering events caused by satellite handover and maintaining reasonably good video quality and smoothness. SARA achieves this by dynamically adjusting video playback speed and assisting ABR algorithms in selecting the ideal bitrate. The overall structure of an ABR system incorporating SARA is depicted in Figure 7. Upon the current playback status and outage event detected by the outage predictor, SARA optimizer returns three values: buffer scalar, throughput scalar, and playback speed. The scalars are forwarded to the ABR algorithm and act as factors influencing the final bitrate decision of the target ABR algorithm but leaving the core bitrate selection mechanism unchanged. This design enables target ABR to be aware of the influence of LSN outage events and requires minimal modifications to existing algorithms.

Figure 8 illustrates an example of how SARA operates alongside other ABR algorithms. When there is an outage event which happens at time 160 with a duration of 2 seconds. In the case of RobustMPC, its throughput predictor failed to anticipate the upcoming network disruption. Consequently, the buffer is entirely depleted after the network outage. In comparison, when SARA is integrated with RobustMPC, it detects the impending outage event in advance and reacts by banking more video content both by slowing down the video playback speed and informing RobustMPC to lower the bitrate. Before the network outage that lasted for 2 seconds, it had already stored more than 3 seconds of video data, which allowed the system to withstand the network outage. Subsequently, SARA carefully increases the playback speed after the outage to reduce the LtB back to the desired level, thus maintaining a low-latency viewing experience.

### 3.2 Problem Formulation

In a typical adaptive video streaming system, video content is divided into $K$ chunks with a fixed duration of $\alpha$ seconds. Each chunk is further encoded into multiple bitrate levels of different bitrates, denoted by $\mathcal{B} = \{b_1, b_2, \ldots, b_M\}$ where $M$ represents the total number of bitrate levels. Let $b_{k,m}$ represent the bitrate selected for the k-th chunk, and $q(b_{k,m})$ represent the quality received by the user

Figure 8: buffer occupation during a network outage.

for the selected bitrate using any video quality metric (e.g., PSNR and VMAF [30]). Let $\xi_k$ represent the average throughput, $C_k$ represent the current buffer occupation in seconds, and $LtB_k$ signify the current LtB in seconds. We denote the predicted outage event at chunk $k$ as a 2-tuple $o_k = (o_k^t, o_k^d)$, where $o_k^t$ is the time remaining until the outage occurs, and $o_k^d$ is the duration of the outage.

For each decision round, our control variables consist of the buffer scalar $s_k^b$, the throughput scalar $s_k^r$, and the playback speed $\beta_k$. The buffer scalar and the throughput scalar range between 0 and 1 aimed to bias the ABR algorithms to select the appropriate bitrate. In addition, the range of $\beta_k$ is constrained between 0.95 and 1.03 to ensure that changes in playback speed remain imperceptible to the viewers, as corroborated by prior research [7]. We denote $\psi_x(\cdot)$ as the generalized optimization target where the inputs are buffer occupancy $C_k$ and/or throughput $\xi_k$, and the output is bitrate $b_{k,m}$. Thus, the selected bitrate can be represented as:

$$b_{k,m} = \begin{cases} \psi_b(s_k^b C_k), & \text{if } \psi_x(\cdot) \text{ is buffer based} \\ \psi_r(s_k^r \xi_k), & \text{if } \psi_x(\cdot) \text{ is throughput based} \\ \psi_h(s_k^b C_k, s_k^r \xi_k), & \text{if } \psi_x(\cdot) \text{ is hybrid or learning network based} \end{cases}$$

Prior to the onset of outage $o_k$, we have a duration of $o_t^k$ video content on the fly that can be downloaded. The maximum number of downloadable chunks prior to this outage is denoted as $\theta_k$. The actual $\theta_k$ is affected by our current throughput $\xi_k$ and the selected bitrate $b_{k,m}$. Additionally, $\theta_k$ is constrained by the number of chunks produced by the broadcaster within the time period $o_t^k$. Thus, we can formulate the number of chunks that can be downloaded as:

$$\theta_k = \min\left\{ \left\lfloor \frac{\xi_k o_t^k}{b_k \alpha} \right\rfloor, \left\lfloor \frac{o_t^k}{\alpha} \right\rfloor \right\}, \forall k \in \mathcal{K}, \quad (1)$$

Given the operational constraint that a video chunk can only be decoded once the entire chunk is fully downloaded, it is necessary to round down the $\theta_k$ value to account for incomplete chunks. Taking into account the predicted outage events during each interval, we apply the following constraint to ensure that the buffer remains healthy after an outage:

$$\frac{1}{\beta_k}(C_k + \theta_k \alpha) - o_k^t - o_k^d \geq \gamma, \forall k \in \mathcal{K}, \quad (2)$$

The first term encapsulates the total available video content for playback before an outage $o_k$ occurs. This is calculated by adding the already buffered video content to the video content that can be downloaded, adjusted by the playback speed. After subtracting $o_k^t$ and $o_k^d$, we get the size of the remaining available buffer. This value

**Algorithm 1** Video Adaption Process using SARA

1: **for** $k$ = 1 to $\mathcal{K}$ **do**
2:      $C_k \leftarrow$ current buffer state
3:      $\xi_k \leftarrow$ throughput measured when downloading chunk $k - 1$
4:      $(o_k^t, o_k^d) \leftarrow Outage\_Predictor()$
5:      $(s_k^b, s_b^r, \beta) \leftarrow SARA\_Optimizer(o_k^t, o_k^d)$
6:      **if** $o_k^t > 0$ **then**
7:          $C_k \leftarrow s_k^b \cdot C_k$
8:          $\xi_k \leftarrow s_k^r \cdot \xi_k$
9:      **end if**
10:     set playback speed to $\beta$
11:     $b_{k,m} \leftarrow ABR\_Algorithm(C_k, \xi_k)$
12:     download chunk $k$ with bitrate $m$, wait until finished
13: **end for**

must be equal to or exceed a predefined safety buffer level threshold $\gamma$ to avoid playback interruptions. The necessity for a larger buffer becomes apparent in practical scenarios where outages can result in prolonged re-establishment times at the application level, as highlighted in Section 2.1. Drawing insight from our measurements, we configure $\gamma$ to be 2 seconds.

If Eq. (2) cannot be satisfied for any possible values of $\beta_k$ and $b_{k,m}$ and the buffer runs dry. This means SARA has insufficient time to prepare for the outage. In this case, the rebuffering duration can be defined as:

$$T_k = \max\{o_k^t + o_k^d + \gamma - \frac{1}{\beta_k}(C_k + \theta_k \alpha), 0\} \quad (3)$$

In light of previous work [1, 10, 11, 33, 38], we adopt the following linear-based QoE metric:

$$Q_k = q(b_{k,m}) - \omega T_k - \rho|b_k - b_{k-1}| - \eta|\beta_k - \beta_{k-1}| - \iota L_k, \forall k \in \mathcal{K}, \quad (4)$$

where $|b_k - b_{k-1}|$ represents the video bitrate smoothess and $|\beta_k - \beta_{k-1}|$ represents the playback speed smoothness. The $L_k$ is live latency, which will be applied when $LtB_k$ is greater than the target latency $LtB_0$. In our experiment, the $LtB_0$ is configured to 3 seconds, as per the Low-Latency live guideline [9]. The expression for $L_k$ is as follows:

$$L_k = \max\{LtB_k - LtB_0, 0\}, \quad (5)$$

$\rho, \eta, \omega$ and $\iota$ are adjustment factors to trade-off the quality benefits and penalties. The objective of the system is to find the appropriate buffer scalar $s_b^k$, the throughput scalar $s_r^k$, and the playback speed $\beta_k$ for each chunk $k$ based on the above constraints, with the aim of maximizing total user QoE, represented as:

$$\max_{s_b^k, s_r^k, \beta_k} \sum_{k=1}^{K} Q_k \quad (6)$$
$$s.t. \quad (2) - (5)$$

Algorithm 1 delineates the operational methodology of SARA when it is integrated with any ABR algorithm. The process begins with SARA conducting a preliminary assessment via the Outage_Predictor module to predict upcoming network outages. Following this prediction, SARA utilizes the outage forecasts to adjust the values of several key parameters: the buffer scalar, throughput scalar, and playback speed. These scalars are strategically applied to the current state of the buffer and the estimated throughput, thereby

**Algorithm 2** SARA Optimizer

1: $\mathcal{N} \leftarrow$ number of iterations
2: $\mathcal{R} \leftarrow$ number of particles
3: $\mathcal{A} \leftarrow$ aggressiveness
4: $w_1, w_2, w_3 \leftarrow$ initialize weights
5: **if** $C_k < o_k^t$ **then**
6:      $O \leftarrow \max\{C_k - o_k^t / C_k, -0.2\}$
7: **end if**
8: **for** $i$ = 1 to $\mathcal{R}$ **do**
9:      $(s_{k,i}^b, s_{k,i}^r, \beta_{k,i}) \leftarrow$ randomly generate within bounds
10:     $v_i^{s_k^b}, v_i^{s_k^r}, v_i^{\beta_k} \leftarrow$ randomly generate between 0 and $\mathcal{A}$
11:     $(p_i^{s_k^b}, p_i^{s_k^r}, p_i^{\beta_k}) \leftarrow (s_{k,i}^b, s_{k,i}^r, \beta_{k,i})$
12:     $QoE_i = QoE(s_{k,i}^b, s_{k,i}^r, \beta_{k,i}) \leftarrow$ calculate using Equation 6
13: **end for**
14: $G^{s_k^b}, G^{s_k^r}, G^{\beta_k} \leftarrow \arg\max\{QoE_i, \ldots, QoE_n\}$
15: **for** $n$ = 1 to $\mathcal{N}$ **do**
16:     **for** $i$ = 1 to $\mathcal{R}$ **do**
17:        $r_1, r_2 \leftarrow$ randomly generate between 0 and 1
18:        $v_i^{s_k^b} \leftarrow w_1 \cdot v_i^{s_k^b} + w_2 \cdot r_1 \cdot (p_i^{s_k^b} - s_k^{b_i}) + w_3 \cdot r_2 \cdot (G^{s_k^b} - s_k^{b_i})$
19:        $s_{k,i}^b \leftarrow \min\{0, s_k^b + v_i^{s_k^b} + O\}$
20:        repeat steps 17-19 for $s_{k,i}^r$ and $\beta_{k,i}$
21:        **if** $QoE(s_{k,i}^b, s_{k,i}^r, \beta_{k,i}) > QoE(p_i^{s_k^b}, p_i^{s_k^r}, p_i^{\beta_k})$ **then**
22:           $(p_i^{s_k^b}, p_i^{s_k^r}, p_i^{\beta_k}) \leftarrow (s_{k,i}^b, s_{k,i}^r, \beta_{k,i})$
23:        **end if**
24:     **end for**
25:     $G^{s_k^b}, G^{s_k^r}, G^{\beta_k} \leftarrow \arg\max\{QoE_i, \ldots, QoE_n\}$
26: **end for**
27: **return** $G^{s_k^b}, G^{s_k^r}, G^{\beta_k}$

modulating the inputs to the ABR algorithm. This modulation is designed to decisively influence the ABR's bitrate selection. In scenarios where no outage is anticipated, and the LtB metric is within optimal ranges, these scalars default to a value of 1. This means that the ABR algorithm operates without external adjustments, maintaining full autonomy over bitrate decisions.

Since both buffer and throughput scalars, as well as playback speed, are continuous variables, identifying the optimal solution can be computationally intensive, rendering it impractical for live streaming contexts. To address this, we have employed a heuristics-based strategy using Particle Swarm Optimization (PSO) [4], as detailed in Algorithm 2. In this algorithm, we spawn $\mathcal{R}$ particles, each representing a potential state vector $(s_{k,i}^b, s_{k,i}^r, \beta_{k,i})$ with random initial values. Each particle is assigned a velocity vector $(v_i^{s_k^b}, v_i^{s_k^r}, v_i^{\beta_k})$ between 0 and $\mathcal{A}$, which dictates the magnitude of exploration within the solution space. The exploration aggressive, denoted by $\mathcal{A}$, determines how aggressively the particles explore the solution space. Furthermore, every particle maintains a record of its optimal state encountered thus far, represented as $(p_i^{s_k^b}, p_i^{s_k^r}, p_i^{\beta_k})$. Additionally, a global vector denoted as $G^{s_k^b}, G^{s_k^r}, G^{\beta_k}$ is kept to track the state vector with the highest $QoE$ observed across all particles so far. During the optimization phase, we update the velocity of each

particle based on its relative displacement from both its local best state and the global best state. This update is moderated by weight factors $w_1$, $w_w$ and $w_3$ to balance exploration and exploitation dynamics and is randomized through coefficients $r_1$ and $r_2$ (uniformly distributed between 0 and 1) to introduce variability in the search process. To further the algorithm's performance, we introduce a variable $O$, which assesses the buffer health relative to upcoming predicted network outages. A higher $O$ value encourages exploration towards lower state values, facilitating faster convergence to more advantageous states. This exploratory process is iterated $N$ times to ascertain the final configurations for the buffer scalar, throughput scalar, and playback speed.

## 4 EVALUATION

### 4.1 Experimental Setup

Considering the dynamics of LSNs and the fact that the exact time and duration of network outages caused by satellite handover are not accessible ahead of time, we conduct evaluations in a simulated environment as widely used in [10, 11, 15, 25, 32] to ensure a fair assessment of SARA in a live streaming environment under LSNs. The simulated network conditions, such as ping and bandwidth, were based on the real-world data we gathered from our measurements on Starlink. The outage $o_k = (o_k^t, o_k^d)$ sampled from Section 2.2 will be embedded into the network trace to emulate the realistic Starlink outages. The video trace utilized in our experiment featured a ten-minute movie from Big Buck Bunny[10]. The movie was processed using the same standard definition encoding as [33], with four bitrates $\mathcal{B} = \{1000, 2500, 5000, 8000\}$ Kbps and a segment duration of 500ms. For the QoE metric in Eq. (4), we adopt two widely used settings as in [14, 25, 33, 38]. The linear quality metric $QoE_{lin}$ with $q(b_k) = b_k/1000$, $\omega = 4.33$, $\rho = 1$, and the log-form quality metric $QoE_{log}$ with $q(b_k) = log(b_k/min(\mathcal{B}))$, $\omega = 2.66$, $\rho = 1$. $\eta$ and $\iota$ are set to $min(\mathcal{B})$ and 1 respectively inspired by [1].

### 4.2 Baseline Algorithms

To evaluate the versatility of SARA, we selected a variety of well-known ABR algorithms. These algorithms represent a broad cross-section of different approaches to adaptive bitrate control, and their diverse strategies and performance characteristics make them an ideal testing ground for SARA's performance and compatibility. We have further modified these ABR algorithms to better align with live streaming scenarios. The chosen ABRs are as follows:

- RobustMPC [38]: Enhances user QoE by using the prediction of future network throughput with the harmonic mean of past throughput.
- Pensieve [25]: Employs reinforcement learning, allowing for adaptation to various environments and QoE metrics.
- BOLA (Buffer Occupancy based Lyapunov Algorithm) [33]: Utilizes Lyapunov optimization to minimize rebuffering and maximize video quality.
- BBA [12]: Decides bitrates based on current buffer capacity and reservoir estimation.

[10]https://peach.blender.org/

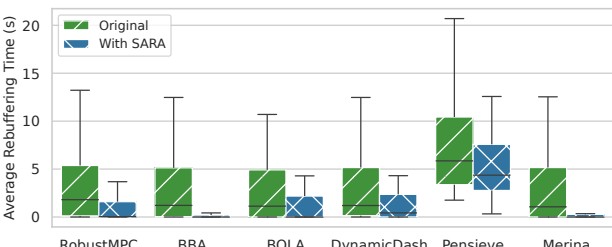

**Figure 9: Average rebuffering time of different ABR algorithms and their integration with SARA.**

- DynamicDash [31]: The default ABR algorithm used by the current standard DASH reference player (4.7.3)[11] which switches between throughput based and buffer based ABR.
- Merina [14]: Utilizes meta reinforcement learning to rapidly adjust its control policy in response to the dynamic changes in network throughput.

### 4.3 SARA Performance Evaluation

We integrate SARA into six previously mentioned ABR algorithms and evaluate their performance. The results, illustrated in Figure 9, highlight SARA's impact in terms of rebuffering times. On average, rebuffering time is reduced by 39.41%, with the BBA achieving an average reduction of 52.26%. It is worth noting that there is a significant standard deviation observed in average rebuffering times. This variability is attributed to the broad range of outage durations in the Starlink network, as discussed in Section 2.2. With network outages lasting anywhere from 0.2 seconds to 23.57 seconds, such a wide spectrum of durations inevitably results in significant fluctuations in average rebuffering times. Moreover, our network traces include a few instances without any network outages, further adding to the observed variance. Figure 10 further shows the impact of SARA on three other metrics across the selected ABR algorithms and their SARA-enhanced versions. SARA results in a slight improvement in LtB of 0.65% and only incurs a negligible reduction in bitrate by 0.13%. Although it leads to a 23.36% increase in the amount of time where the playback speed differs from 1x, compared to the original ABR algorithms, SARA only alters the playback speed within a narrow range from 0.95x to 1.03x, which is typically imperceptible to viewers in practice [1].

We also notice that Pensieve appears worse compared to other ABR algorithms. Further analysis reveals that Pensieve tends to respond slowly to outage events. It often reduces the bitrates with a delay, by which time the bandwidth may have already recovered, resulting in reduced bandwidth efficiency and an increase in rebuffering time. Although Pensieve does not perform optimally under LSN conditions, the integration of SARA still improves its performance, resulting in an average of 33% less rebuffering time.

Figure 11 and Figure 12 present the performance evaluation in the form of CDFs for each ABR algorithm and their integration with SARA among two QoE metrics. The results show that SARA either matches or surpasses the performance of the base ABR algorithm that it is incorporated into. For example, for Merina, 75% of $QoE_{log}$ values are above 0.8, while when SARA is integrated with it, 80% of QoE values exceed 0.97. In essence, SARA's ability

[11]https://reference.dashif.org/dash.js/

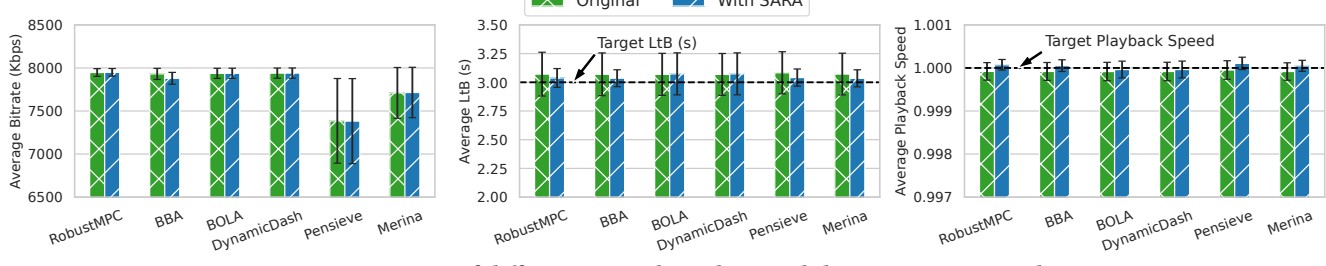

Figure 10: QoE metrics of different ABR algorithms and their integration with SARA.

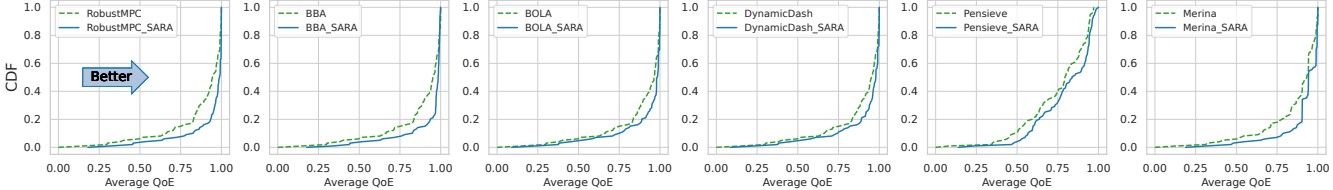

Figure 11: CDF of $QoE_{lin}$ metrics of different ABR algorithms and their integration with SARA.

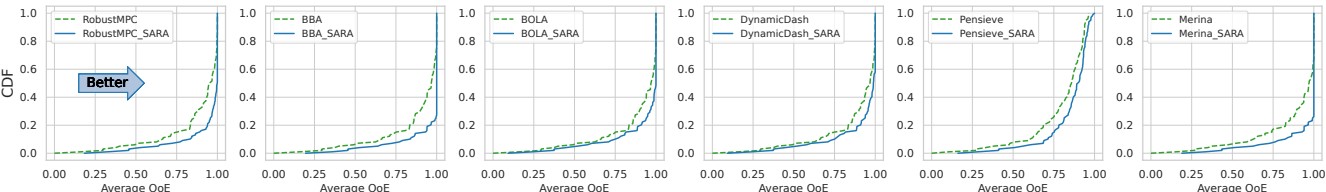

Figure 12: CDF of $QoE_{log}$ metrics of different ABR algorithms and their integration with SARA.

Table 2: Performance of the outage predictor with various configurations

| Input Length | Prediction Length | MSE | MAE |
|---|---|---|---|
| 300 | 30 | 0.519 | 0.107 |
| | 60 | 0.521 | 0.113 |
| | 120 | 0.542 | 0.121 |
| **600** | 30 | 0.517 | 0.107 |
| | 60 | 0.523 | 0.116 |
| | **120** | **0.563** | **0.119** |
| 900 | 30 | 0.524 | 0.109 |
| | 60 | 0.567 | 0.120 |
| | 120 | 0.619 | 0.129 |

to predict outages, dynamically adjust playback speed, and assist ABRs in selecting the ideal bitrate forms an efficient strategy that significantly enhances these ABR algorithms' performance, thereby improving the live streaming experiences in LSN environments.

## 5 FURTHER DISCUSSION: OUTAGE PREDICTOR

A key component of SARA is the "Outage Predictor" which can forecast upcoming network outages from satellite handovers. However, predicting network outages presents a significant challenge due to the black box design of satellite operators' system architecture, such as Starlink [20, 24]. Essential details such as the strategies for satellite handovers, network routing, and satellite system load remain proprietary to the general public [29, 35, 41]. Fortunately, the orbital positions of Starlink satellites are accessible publicly in the Two-Line Element (TLE) format from various sources. This

accessibility enables the calculation of the satellites' positions relative to the UE at any moment. Moreover, the orbital paths of the satellites exhibit consistent, timed characteristics, recurring at fixed intervals. This regularity positions the problem well for analysis using time-sequenced modelling techniques. Furthermore, recent advancements in transformers offer promising solutions for analyzing and predicting time-sequenced data [36]. Therefore, one possible further enhancement on SARA is to utilize *Informer*, a state-of-the-art transformer-based approach designed for efficient long-sequence time-series forecasting in the design of the outage predictor [42]. In this section, we further discuss our initial efforts in this direction.

In our informer based design, the training data incorporates the measurement data outlined in Section 2 and TLEs from CelesTrak[12]. Since we do not know which satellite Starlink will switch to after each handover period, all satellites that have an angle of elevation higher than $25°$, along with their distance to the UE, are included in the training data. To further improve the accuracy of outage predictor, we also integrate weather data sourced from the *Weather API* provided by OpenWeather[13], acknowledging the impact of weather conditions on the performance of the Starlink network, as highlighted in previous studies [24, 27, 41].

We train the Informer network under a range of configurations, varying both the input length and the prediction length. For instance, an input length of 300 and a prediction length of 30 means the model will analyze the latest 5 minutes of data to forecast network outages in the upcoming 30 seconds. An extended prediction

---

[12]https://celestrak.org/NORAD/Elements/table.php?GROUP=starlink
[13]https://openweathermap.org/current

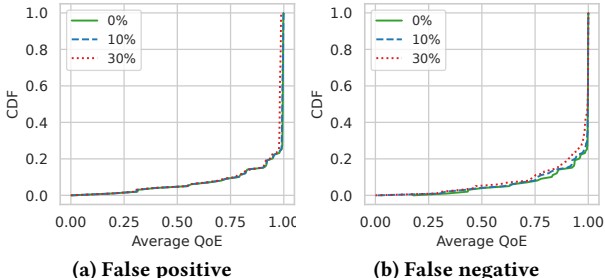

(a) False positive      (b) False negative

Figure 13: CDF of $QoE_{lin}$ with BBA_SARA under various levels of false positive and false negative outage predictions.

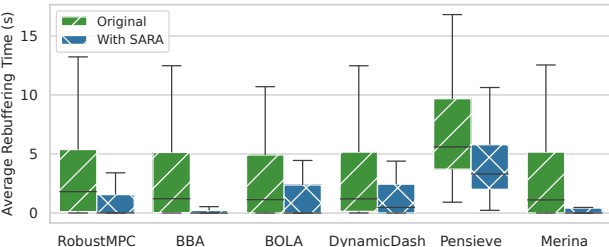

Figure 14: Average rebuffering time of different ABR algorithms and their integration with SARA using a transformer based outage predictor.

length equips SARA with a longer preparation window ahead of network outages, potentially enhancing its resilience to prolonged disruptions. On the other hand, augmenting the input length tends to improve performance due to the richer contextual data provided to the model, albeit at the cost of increased inference times. The impact of these variations on the model's accuracy is summarized in Table 2, which presents the results in terms of Mean Squared Error (MSE) and Maximum Absolute Error (MAE).

We further performed subsequent analysis focusing on evaluating how the accuracy of outage predictions influences overall user QoE. Figure 13 delineates the comparative performance between BBA and BBA_SARA under various scenarios characterized by differing rates of false positives and false negatives in outage predictions. The results indicate that false negative predictions exert a more significant detrimental effect on user QoE. This observed impact is directly linked to false negatives precipitating unforeseen video rebuffering events, which significantly compromise the viewing experience. Conversely, a temporary reduction in bitrate, if only for a brief duration, tends to affect the overall QoE for viewers minimally. In response to these insights, we adjust our outage predictor's loss function during training to impose a higher penalty for false negatives, enhancing its sensitivity to potential outages. After fine-tuning, the optimal configuration for the model was determined to be an input length of 600 seconds and a prediction length of 120 seconds. The outage predictor achieved an overall accuracy and recall rate of 79.43% and 38.23%, respectively. It is important to note that this model was developed with a constrained dataset. In practice, satellite operators such as SpaceX Starlink and Amazon Kuiper have access to more comprehensive data, enabling the development of predictors with even superior performance metrics. Nevertheless, our model still offers a versatile solution that is simultaneously applicable across various satellite operators, including SpaceX Starlink and Amazon Kuiper and future LSN operators. Despite the limited data, our model still effectively reduces rebuffering events, as illustrated in Figure 14.

## 6 RELATED WORKS

The domain of LSNs has been extensively explored in recent research, with a significant focus on the Starlink network, emphasizing comprehensive end-to-end network performance evaluations. These studies have delved into environmental influences on network performance, such as terrain and weather conditions [17, 24, 27, 41], affirming Starlink's applicability across a spectrum of applications while underscoring the challenges inherent to LSNs,

notably the frequency of satellite handovers. Further investigations into satellite handovers within LSNs have unveiled the deployment of a global scheduler by Starlink, which executes handover decisions at 15-second intervals [29, 35]. Concurrently, advancements in simulation tools have deepened our understanding of LSN operational conditions [18, 19], enhancing the fidelity of network models. Efforts to improve lower-level TCP protocols have also been pursued, aiming to enhance overall LSN performance [5].

In the context of video streaming services, significant advancements have been made in the development of ABR algorithms that adeptly respond to fluctuating network conditions. These include buffer based approaches [12, 33], throughput based approaches [13, 34] and a hybrid of both [31, 38], which are simple yet effective. Additionally, innovative learning-based ABR algorithms, notably Pensieve and Comyco, are capable of directly optimizing QoE without the necessity for iterative computation [8, 14, 15, 20, 25, 37, 39]. ABR algorithms such as LoL and LoL+ have been tailored specifically for low latency in live streaming platforms [1, 6, 16, 21, 26], integrating bitrate adaptation with heuristic based playback speed control. In parallel, various studies have focused on minimizing hardware load and enhancing power efficiency, marking significant strides towards more sustainable streaming technology [23, 43].

## 7 CONCLUSION AND FUTURE WORKS

In this paper, we examined the challenges of live streaming over LSNs, notably the impact of frequent satellite handovers on live streaming services. Our investigation revealed that existing ABR algorithms, which perform well in terrestrial network environments, struggle to maintain consistent streaming quality in the dynamic conditions characteristic of LSNs. We introduced SARA, a versatile and lightweight middleware solution designed to enhance the adaptability of existing ABR algorithms in LSN contexts by feeding ABR algorithms with specific network characteristics unique to LSNs. Our evaluation shows that SARA significantly reduced rebuffering times by approximately 39.41% while only incurring a neglectable loss in an average bitrate of 0.65%, demonstrating its efficacy in optimizing live streaming services in satellite networks.

LSN services are rapidly evolving and continue to present many fascinating and challenging areas in need of exploration. An area of particular interest is the realm of realtime communication and asynchronous interaction, which has not yet been extensively investigated. Our future work will delve into these uncharted territories, seeking to optimize the performance of LSNs in facilitating seamless, realtime interactions.

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
