# OpenReview forum: "Toward Robust Live Streaming over LEO Satellite Constellations: Measurement, Analysis, and Handover-Aware Adaptation"
_acmmm.org/ACMMM/2024/Conference — MM2024 Poster_

### Official Review · Reviewer_XS3d · 2024-05-13

**Rating:** 4
**Confidence:** 3

**Summary:**

The paper introduces a "middleware" that aims to predict regular handover interruptions of low earth orbit satellite network connections
and dynamically adjust the playback speed of video streaming sessions to avoid video playback stalling during the interruptions. The authors conducted simulations to show the benefit of the design in comparison with multiple standard ABR techniques.

**Strengths:**

The paper has a clear focus and in-depth analysis of the issues faced by low earth orbit satellite network (LSN). Real data have been captured to model the network characteristics and a detailed design is in place to balance bitrate, playback speed, and user experience under different network conditions. A simulation has been conducted to evaluate the potential benefits of the proposed design and a comparison with multiple standard ABR is made. Overall the paper is well written and relevant to the conference.

**Limitations:**

A few things related to design and evaluation require further clarification.
1. The paper focuses on live streaming and uses Twitch streaming as the use case. It is clear that the live content has a few seconds of buffer / delay already to counter network issues. If most outages only last a couple of seconds, why can't you just increase the initial buffer length from 3 seconds to 5 seconds or even longer? At the end of the day, it's not interactive content anyway. And a longer buffer would be a much easier solution.
2. The definition of outage in this paper is confusing. Do you mean network outage (loss of connection) or outage observed on user devices (frame freezing or skipping)? In Section 3.1 it says "...the buffer is entirely depleted after the network outage" but in Figure 8 the buffer depletion (green line)  drops to 0 before the red line (network outage). My understanding is that the network outage should have appeared a lot earlier and the red line is when the video playback stalled.
3. The experiment uses Big Buck Bunny streaming. Although it's a common practice to do such a simulation, I am not sure if the setup is correct to simulate live streaming. I feel the authors are doing on-demand streaming instead. The difference is that with on-demand streaming, users will still get all the content after stalling while for live streaming users would have missed some content after the playback restarts after stalling. Perhaps Twitch has a server-side buffer but it is not clear to me. I guess that even if you stretch the current buffer longer by slowing down the playback, you will still get a huge frame skipping after a network outage due to the missing content (providing that you are doing proper live streaming and not VOD). Please clarify this in the paper.
4. It claims that the playback speed change to between 0.95 and 1.03 is not noticeable. Does this apply to audio or only video? Please clarify.

**Suitability:**

3

---

### Official Review · Reviewer_BFd9 · 2024-05-22

**Rating:** 4
**Confidence:** 2

**Summary:**

The paper proposes an enhancement to Adaptive Bitrate (ABR) algorithms for use in networks that include LEO satellite networks. The key idea is to use knowledge or inference of upcoming satellite handovers to modify the ABR algorithm being used, to reduce the time spent rebuffering live streams due to the abrupt nature of these handovers. The proposed approach (SARA) is described in detail, and evaluated by simulation using real-world data on a number of commonly-used ABR algorithms. The results indicate that SARA can reduce rebuffering times without having much impact on average bitrates or latencies.

**Strengths:**

Providing a way for any ABR algorithm to be aware of satellite handovers and take them into account with little disturbing of the user is a sensible approach.

SARA is described and justified, as well as how it could be implemented in practice using a particle swarm optimization heuristic approach.

Basing the analysis of the impact of satellite network handovers, as well as the data for the simulation experiments, on the Starlink system has benefits in terms of realism. Several well-known ABR algorithms of different types are used to explore the impact of SARA on their performance.

**Limitations:**

Reducing the rebuffering time is likely to be beneficial to all users, but the number of rebufferings may also be an important element of a user's QoE, yet this is not captured in the proposed approach. In other words, having the same number of interruptions which are on average shorter may not be as significant a benefit (at least for some users) as it appears in this paper.

SARA adopts a particular form of QoE (equation 4) but this may not be a good match to an individual user's QoE, which means the objective of the algorithm for that user might not be accurate for them. Also, equation 6 shows that all chunks contribute equally to a user's QoE, but some chunks may be more important than others (depending on the stream content), and this doesn't capture the recency effect where a user's instantaneous QoE depends on their most recent experience with the stream.

The reliance on Starlink for insight into satellite network behavior and experimental data, while bringing some benefits, also raises the question of how applicable this proposed approach would be to other satellite networks.

The computational overhead per UE seems high e.g. running Algorithm 2 with R particles; continuously running an outage predictor. This overhead is not addressed in the paper.

If accepted, some minor changes or corrections are needed. For example:

"ultimately leads to a rebuffering event lasting for 1.4 seconds" [Figure 3] - should be 14 seconds

"with a probability of 2.73%" - probability should always be specified in the range [0,1]

page 4, several instances of (o^k)_t which should be (o^t)_k
(subscript and superscript were incorrectly swapped )

page 6, there is a word or phrase missing from "To further the algorithm's performance"

"remain proprietary to the general public" should be rewritten to something like "remain proprietary and therefore hidden from the general public"

**Suitability:**

3

---

### Official Review · Reviewer_QXM5 · 2024-05-23

**Rating:** 3
**Confidence:** 3

**Summary:**

This paper discusses the challenges and solutions for live streaming over Low Earth Orbit Satellite Networks (LSNs). The authors highlight that frequent satellite handovers in LSNs, like SpaceX’s Starlink, cause network outages that disrupt the streaming experience. To address these challenges, the authors introduce Satellite-Aware Rate Adaptation (SARA), a middleware designed to enhance live streaming over LSNs. SARA works with Adaptive Bitrate (ABR) algorithms to minimize rebuffering events during satellite handovers. The authors conducted a measurement study on Starlink’s network, revealing that existing ABR algorithms struggle with the variable network conditions of LSNs, leading to frequent rebuffering events. The simulation results of SARA show a reduction in rebuffering time and a slight improvement in latency.

**Strengths:**

1.	The topic is interesting. LEO satellite networks are very popular, and LEO mega-constellations for live video streaming are widely adopted.

2.	The background is clearly presented. The authors provide adequate information about the topic.

3.	The measurement study is good for the readers to understand the rebuffering issue.

**Limitations:**

1.	Satellite handovers will introduce rebuffering issues in live video streaming. However, compared to the proposed method of mitigating the impact of satellite handovers on rebuffering, why don’t you focus on ways of minimizing the number of satellite handovers?

2.	According to the measurement study, an average of 1.04 rebuffering events occur per hour with LEO satellite networks. The frequency of rebuffering events is low.

3.	The authors state an approximately 80% chance of experiencing an outage within a 60-minute interval. However, it is said that satellite handover occurs every 15 seconds. This means the opportunity for handovers leading to network outages could be more manageable.  I wondered in what specific cases the handover would lead to a network outage.

4.	Also, the authors claim that the frequency of outages experiences a dramatic increase from 15:00 to 18:00, peaking at 20:00 (Internet rush hour). The major issue leading to the network outage is not the satellite handovers.

5.	When SARA is integrated with RobustMPC, how does it detect the impending outage event in advance?

6.	The authors employ a heuristic-based approach using PSO. Why is PSO chosen?

7.	The SARA is based on the predicted network outages to mitigate rebuffering. Predicting network outages is an important issue, but I can hardly find a solution in this paper.

8.	The constellation setting used in your simulations should also be clearly described.

9.	Also, how long does it take to predict the handovers compared to the real-world handover interval (15 seconds)?

10.	Using predicted handover data exhibits comparable performance to using real handover data. How does it work? Besides, in some cases (Pensieve), using predicted handover data even outperforms real handover data.

**Suitability:**

3

---

### Meta-Review · Area_Chair_KXF7 · 2024-07-04

**Recommendation:** Accept (Poster)
**Confidence:** 5

**Metareview:**

Pros:

- Well-studied background and SotA analysis

Cons

- very general and confusing definition of "outage."
- Simulation instead of real implementation
- Clarity in the evaluation setups and experimental results